

# The largest European theropod dinosaurs: remains of a gigantic megalosaurid and giant theropod tracks from the Kimmeridgian of Asturias, Spain

Oliver W.M. Rauhut[1,2,3], Laura Piñuela[4], Diego Castanera[1,2], José-Carlos García-Ramos[4] and Irene Sánchez Cela[4]

[1] SNSB, Bayerische Staatssammlung für Paläontologie und Geologie, Munich, Germany
[2] GeoBioCenter, Ludwig-Maximilians-University, Munich, Germany
[3] Department for Earth and Environmental Sciences, Ludwig-Maximilians-University, Munich, Germany
[4] Museo del Jurásico de Asturias, Colunga, Spain

Corresponding author
Oliver W.M. Rauhut,
o.rauhut@lrz.uni-muenchen.de

## ABSTRACT

The Kimmeridgian Vega, Tereñes and Lastres formations of Asturias have yielded a rich vertebrate fauna, represented by both abundant tracks and osteological remains. However, skeletal remains of theropod dinosaurs are rare, and the diversity of theropod tracks has only partially been documented in the literature. Here we describe the only non-dental osteological theropod remain recovered so far, an isolated anterior caudal vertebra, as well as the largest theropod tracks found. The caudal vertebra can be shown to represent a megalosaurine megalosaurid and represents the largest theropod skeletal remain described from Europe so far. The tracks are also amongst the largest theropod footprints reported from any setting and can be assigned to two different morphotypes, one being characterized by its robustness and a weak mesaxony, and the other characterized by a strong mesaxony, representing a more gracile trackmaker. We discuss the recently proposed distinction between robust and gracile large to giant theropod tracks and their possible trackmakers during the Late Jurassic-Berriasian. In the absence of complete pedal skeletons of most basal tetanurans, the identity of the maker of Jurassic giant theropod tracks is difficult to establish. However, the notable robustness of megalosaurine megalosaurids fits well with the described robust morphotypes, whereas more slender large theropod tracks might have been made by a variety of basal tetanurans, including allosaurids, metriocanthosaurids or afrovenatorine megalosaurids, or even exceptionally large ceratosaurs. Concerning osteological remains of large theropods from the Late Jurassic of Europe, megalosaurids seem to be more abundant than previously recognized and occur in basically all Jurassic deposits where theropod remains have been found, whereas allosauroids seem to be represented by allosaurids in Western Europe and metriacanthosaurids in more eastern areas. Short-term fluctuations in sea level might have allowed exchange of large theropods between the islands that constituted Europe during the Late Jurassic.

## INTRODUCTION

In the Late Jurassic, Europe was an assemblage of numerous smaller to large islands, separated by shallow epicontinental seas (*Cosentino et al., 2010*: Fig. 7). Apart from the Fennoscandian shield, representing the largest continental mass in north-eastern Europe, larger landmasses included, from east to west, the Bohemian Massif (approximately where the Czech Republic lies today), the London–Brabant Massif and the Rhenian Isle (extending from the area around London to the lower Rhine embayment), the Massif Central (south-central France), the Armorican Massif (mainly the Bretagne today), the Irish Massif in the north-west, and the Iberian Massif (Portugal and parts of western Spain). During parts of the Late Jurassic, the London–Brabant–Rhenian Massif and the Bohemian Massif might have been connected in the north, and the Armorican Massif might have partially had a connection with the Massif Central (*Thierry et al., 2000*; *Meyer, 2012*). All of these landmasses certainly possessed a fauna of terrestrial vertebrates, but little is still known about many of these faunas.

Apart from the record of the Iberian Peninsula, in which abundant terrestrial vertebrates are mainly found in Late Jurassic terrestrial to transitional sediments of the Lusitanian (see *Mocho et al., 2017*, and references therein), Maestrazgo, and South Iberian basins (*Royo-Torres et al., 2009*; *Aurell et al., 2016*; *Campos-Soto et al., 2017*), most records of Late Jurassic dinosaurs from Europe come from shallow marine sediments, such as the famous lithographic limestones of southern Germany (*Rauhut & Tischlinger, 2015*; *Tischlinger, Göhlich & Rauhut, 2015*), the Upper Oxford Clay and Kimmeridge Clay of England (see *Benson, 2008a*; *Benson & Barrett, 2009*; *Barrett, Benson & Upchurch, 2010*; *Carrano, Benson & Sampson, 2012*), the marine carbonates at Oker, Germany (*Sander et al., 2006*), the Reuchenette Formation of Switzerland (*Meyer & Thüring, 2003*), the laminated limestones of Canjuers (*Peyer, 2006*), or the Calcaire de Cleval Formation in eastern France (*Mannion, Allain & Moine, 2017*). Interestingly, the sparse evidence from these more eastern occurrences seems to indicate some differences with the fauna from western Iberia. Whereas the latter fauna is closely comparable to the contemporaneous fauna of the Morrison Formation of western North America (*Mateus, 2006*), with even several shared genera being present (*Pérez-Moreno et al., 1999*; *Antunes & Mateus, 2003*; *Escaso et al., 2007*; *Malafaia et al., 2007*, *2015*, *2017a*; *Hendrickx & Mateus, 2014*), at least the theropod fauna from more eastern European localities seems to show some Asian influence, with the metricanthosaurid *Metriacanthosaurus* from the Oxfordian of England (*Huene, 1926*; *Walker, 1964*; *Carrano, Benson & Sampson, 2012*), possible metriacanthosaurid teeth in the Kimmeridgian of northern Germany (*Gerke & Wings, 2016*), and compsognathid and paravian theropods from the Kimmeridgian–Tithonian of the Solnhofen Archipelago (*Ostrom, 1978*; *Wellnhofer, 2008*; *Tischlinger, Göhlich & Rauhut, 2015*; *Foth & Rauhut, 2017*).

Further evidence on the Late Jurassic dinosaur fauna from Europe comes from dinosaur tracksites. Abundant dinosaur tracks are known from the Iberian Peninsula, from different sites within the Lusitanian Basin (*Santos, Moratalla & Royo-Torres, 2009*; *Mateus & Milàn, 2010*), the Villar de Arzobispo Formation of Teruel Province in

Spain (*Canudo et al., 2005*; *Aurell et al., 2016*; *Campos-Soto et al., 2017*), and from the 'dinosaur coast' of Asturias, Spain (*García-Ramos, Piñuela & Lires, 2006*; *Piñuela Suárez, 2015*). Tracksites are also known from the Late Jurassic of France (*Mazin et al., 1997*; *Mazin, Hantzpergue & Pouech, 2016*; *Mazin, Hantzpergue & Olivier, 2017*; *Moreau et al., 2017*), Germany (*Kaever & Lapparent, 1974*; *Diedrich, 2011*; *Lallensack et al., 2015*), Italy (*Conti et al., 2005*), and Poland (*Gierlinski & Niedźwiedzki, 2002*; *Gierlinski, Niedźwiedzki & Nowacki, 2009*), but the largest Late Jurassic track bearing area is certainly that of the Jura mountains of Switzerland (*Marty et al., 2007*, *2017*; *Razzolini et al., 2017*; *Castanera et al., 2018*). Although the identification of theropod tracks to certain clades remains problematic (see also below), these occurrences can give important insights into theropod diversity and community structure.

Apart from the abundant record from the Lusitanian, South Iberian and Maestrazgo Basins, Late Jurassic dinosaur remains, both body fossils and tracks, have also been reported from the Kimmeridgian Vega, Tereñes, and Lastres Formations of Asturias, Spain (*García-Ramos, Piñuela & Lires, 2006*). In the Late Jurassic, Asturias lay between the Lusitanian Basin and the Armorican Massif, either as part of smaller islands (*Cosentino et al., 2010*), or as part of the Iberian Massif (*Thierry et al., 2000*), and its fauna is thus of great interest for understanding European Late Jurassic dinosaur biogeography. Dinosaurs from these units, principally from the Vega and Lastres formations, include mainly ornithischians, with stegosaurs (*Ruiz-Omeñaca et al., 2009a*, *2013*), and ornithopods (*Ortega et al., 2006*; *Ruiz-Omeñaca, Piñuela & García-Ramos, 2007*, *2009b*, *2010*, *2012*) having been reported. Sauropods are rare and include remains of a turiasaur (*Canudo et al., 2010*) and a diplodocid (*Ruiz-Omeñaca, Piñuela & García-Ramos, 2008*). Theropod remains are also rare and consist mainly of isolated teeth (*Canudo & Ruiz-Omeñaca, 2003*; *Ruiz-Omeñaca et al., 2009c*). The only skeletal remain of a theropod is a large anterior caudal vertebra, which was briefly described by *Martínez et al. (2000)* and referred to an unspecified ceratosaur (see also *Canudo & Ruiz-Omeñaca, 2003*). This specimen, which is remarkable for its extremely large size, is re-evaluated here. Furthermore, the Kimmeridgian of Asturias has yielded a rich dinosaur track record (*García-Ramos, Piñuela & Lires, 2006*; *Milàn et al., 2006*; *Avanzini, Piñuela & García-Ramos, 2008*, *2012*; *Lockley et al., 2008*; *Piñuela Suárez, 2015*; *Castanera, Piñuela & García-Ramos, 2016*; *Piñuela et al., 2016*), including isolated tracks of giant theropods (*Piñuela Suárez, 2015*), which are also documented here.

## Geological setting

The main and best-exposed Jurassic outcrops in the Asturias region extend along the sea cliffs between Gijón and Ribadesella localities (Fig. 1). The Jurassic rocks in the eastern part of Asturias overlie diverse Variscan and Permian–Triassic units and can be grouped into two main lithologically and environmentally characterized units. The lower one is predominantly made up of carbonate rocks of littoral-evaporitic (Gijón Formation) and open marine origin (Rodiles Formation). The upper unit mainly comprises siliciclastic rocks of fluvial (Vega Formation), restricted marine (shelf lagoon), and

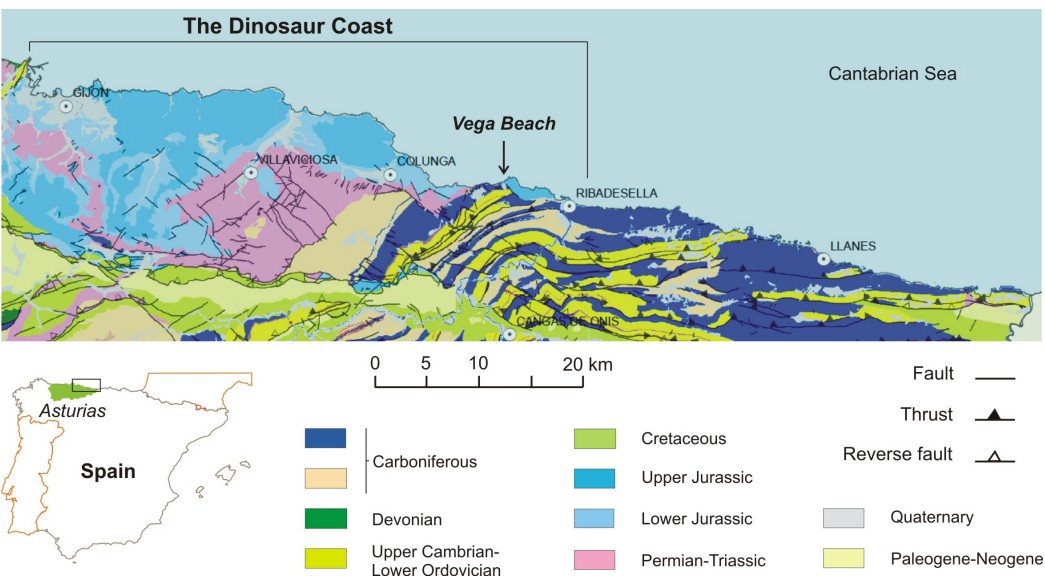

**Figure 1 Geological map of the eastern Asturian sector, including the location of Vega beach (Ribadesella).** Modified after *Merino Tomé, Suárez Rodríguez & Alonso (2013)*.

coastal (fluvial-dominated lagoonal deltas) origin, respectively represented by the Tereñes and Lastres formations (Fig. 2A).

The Vega Formation, with an estimated thickness of 150 m, consists of alternating white, pale grey and reddish sandstones, and red mudstones with several sporadic conglomeratic beds typically arranged in minor finnig-upward cycles within a major cycle of the same character (Fig. 2B). These rocks represent fluvial deposits formed by ephemeral and highly sinuous streams separated by extensive floodplains on which calcareous palaeosols (calcretes) developed (*García-Ramos et al., 2010a*; *Arenas, Piñuela & García-Ramos, 2015*). Based on datations with ostracods and pollen and spores, the age of the Vega Formation is probably Kimmeridgian (*Schudack & Schudack, 2002*; *Barrón, 2010*). The climate during sediment deposition represents warm and semi-arid conditions with a strongly seasonal precipitation regime, as indicated by the local presence of gypsum crystals and veins, the palynological composition (*Barrón, 2010*) and the most frequent palaeosol varieties (*Gutierrez & Sheldon, 2012*).

Fossil prospecting in the Vega Formation type locality, along the coast 6 km west of Ribadesella town (Fig. 1), yielded the theropod caudal vertebra documented in this study. The fossil bone occurred in a 0.65 m thick grey bed of polygenic calcareous microconglomerate (see asterisk in Fig. 2B), which includes mainly carbonate clasts from underlying marine Jurassic units (Gijón and Rodiles formations), together with intraformational limestone and lutitic fragments from the Vega Formation. The calcareous microconglomerate passes upwards to a cross-bedded sandstone. Both lithologies are arranged in at least two finning-upwards channelised levels, showing rapid lateral variations in both thickness and grain-sizes.

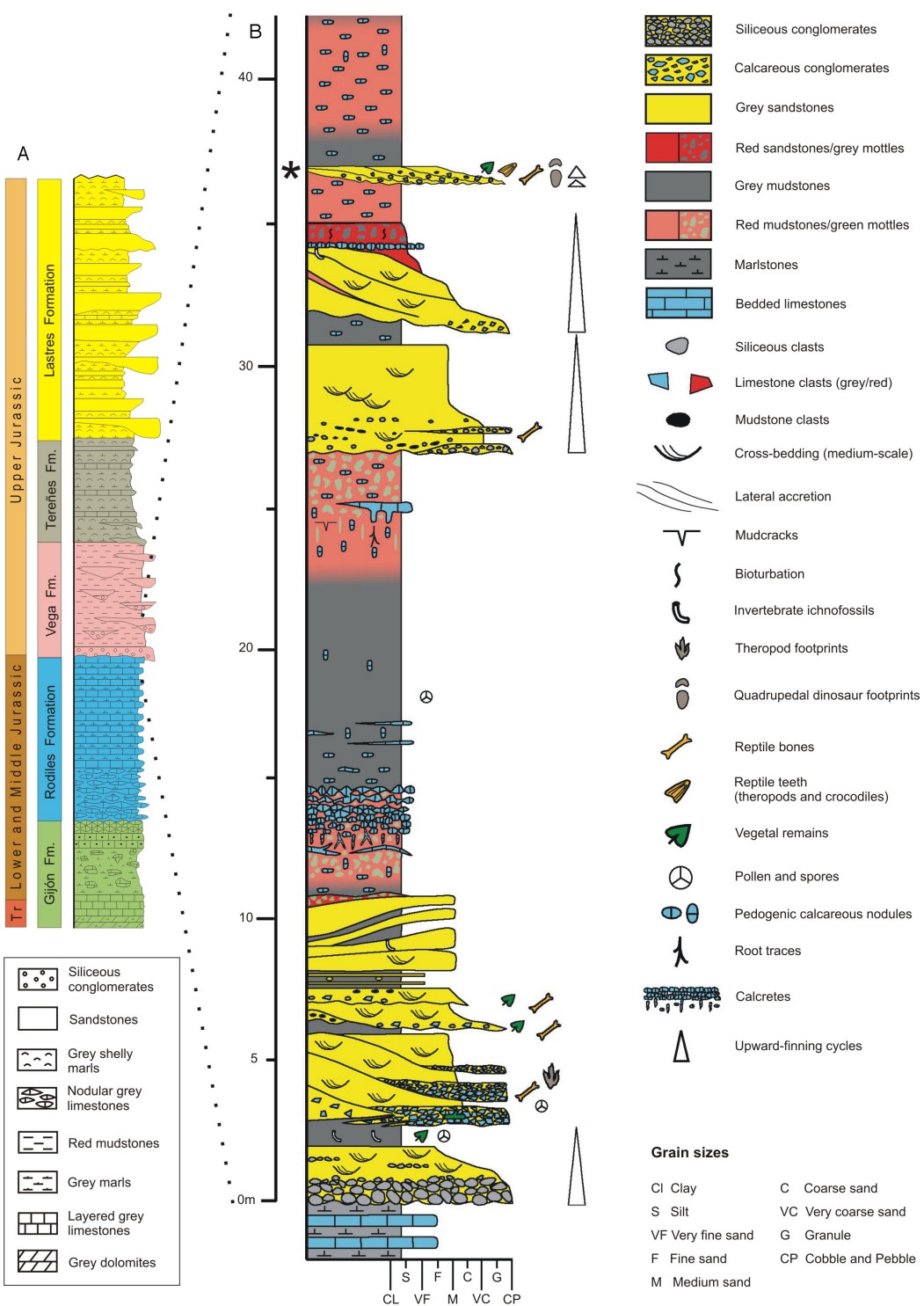

**Figure 2  Geology of the Asturian Jurassic.** (A) General stratigraphic log of the Asturian Jurassic along the Tazones-Ribadesella sector. Not to scale. Modified after *García-Ramos, Piñuela & Rodríguez-Tovar (2011)*. (B) Detailed log of the lower part of the Vega Formation (after *García-Ramos, Aramburu & Piñuela, 2010c*). The level where the vertebra was found is indicated by an asterisk.

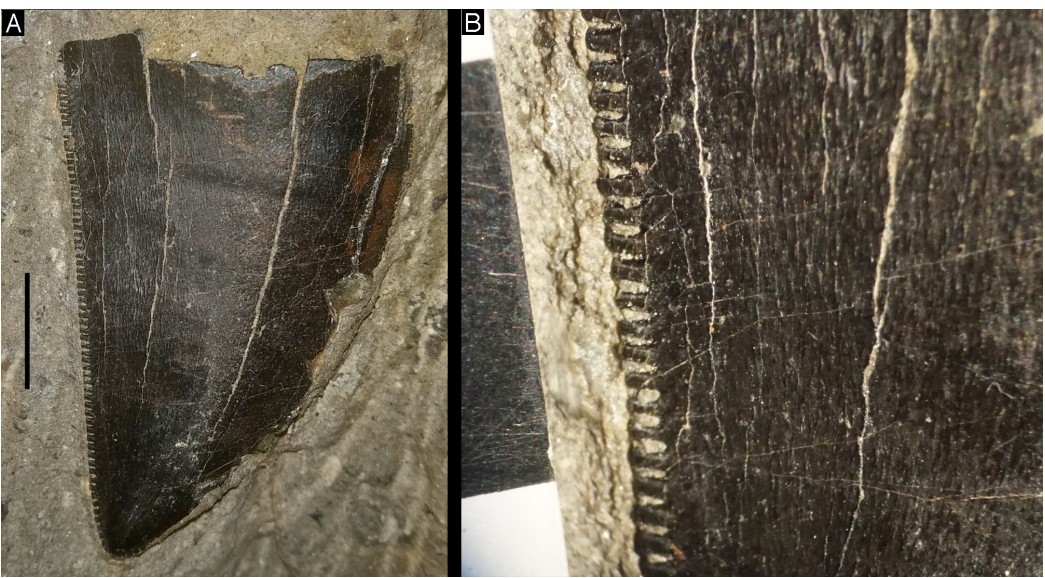

**Figure 3  Tip of a large megalosaurid tooth from the Vega Formation.** (A) General view in lingual or labial view. (B) Detail of distal serrations and anastomosing enamel ornamentation. Scale bars are 10 mm. Photographs by Oliver Rauhut and Diego Castanera.                      

The vertebrate bone bed represents an amalgamation of small lenticular channels (scours) showing several episodes of lateral accretion. Their origin is related to channelised flows produced by extreme flooding events associated with heavy rainfall periods. These high discharge processes are probably supplied by the rapid recharge of water springs from an uppermost Triassic-Lower Jurassic rock aquifer emerging from a nearby fault-controlled calcareous relief located to the south (*García-Ramos et al., 2010a*; *Arenas, Piñuela & García-Ramos, 2015*; *Lozano et al., 2016*).

A tip of a large theropod tooth (MUJA-1226) from the same level as the vertebra described here was reported by *Martínez et al. (2000)* and described in more detail by *Ruiz-Omeñaca et al. (2009c)*. This crown tip is strongly labiolingually compressed, shows centrally placed, serrated carinae, mesiodistally long, rectangular denticles, antapically directed interdenticular sulci, and an anastomosing enamel texture (Fig. 3; see *Ruiz-Omeñaca et al., 2009c*). All of these characters are found in megalosaurid teeth, such as teeth of *Torvosaurus* (*Hendrickx, Mateus & Araújo, 2015*), so this specimen most probably represents a megalosaurid. A smaller theropod tooth was also found in this locality (MUJA-1018; *Ruiz-Omeñaca et al., 2009c*). The same level also included some small oncoids, vegetal remains, turtle fragments, crocodile teeth (*Ruiz-Omeñaca, 2010*), and a sauropod caudal vertebrae (MUJA-0650), as well as poorly-preserved quadrupedal dinosaur footprints, which have not been mentioned or described in the literature so far.

The Lastres Formation is about 400 m thick unit and consists of grey sandstones, lutites, and marls with occasional conglomeratic levels (Fig. 2A). The depositional environment was characterized by a succession of fluvial-dominated lagoonal deltas. The main deposits include prodelta, crevasse-splay, levee, distributary channel, delta front, interdistributary bay, and delta-abandonment facies (*Avanzini et al., 2005*; *García-Ramos,*

*Piñuela & Lires, 2006*; *García-Ramos, Piñuela & Aramburu, 2010b*). Within the Lastres Formation, several short-term transgressive events are recorded by muddy and calcareous laterally extensive shell beds with abundant brackish-water bivalves and gastropods. This formation has provided numerous tracks, not only belonging to dinosaurs, but also to pterosaurs, crocodiles, turtles, and lizards (*García-Ramos, Piñuela & Lires, 2006*; *Piñuela Suárez, 2015*). The footprints here studied were found as loose and isolated sandstone casts on the sea cliffs, thus no precise descriptions of the levels are provided, but most of the Lastres Formation theropod tracks are related to crevasse-splay facies.

## DESCRIPTION

### Osteological remains

The vertebra MUJA-1913 is a large anterior caudal vertebra that has most of the centrum and the base of the neural arch preserved (Fig. 4); the zygapophyses, neural spine and most of the transverse processes are missing. The centrum is notably robust and amphi-platycoelous, with the articular surfaces being oval in outline and slightly higher than wide. The anterior articular surface has suffered from erosion, so that its exact size and morphology cannot be established, but the posterior articular surface is only slightly concave and only slightly higher (c. 150 mm) than wide (c. 140–145 mm as reconstructed; the right rim is eroded). In lateral view, the posterior articular surface is notably offset ventrally in respect to the anterior surface (Fig. 4A). The length of the centrum as preserved is c. 140 mm, but approximately 10 mm might be missing anteriorly, so that the centrum was approximately as high as long. In ventral view, the centrum is moderately constricted to a minimal width of c. 90 mm between the articular ends. Ventrally, a broad, but shallow ventral groove is present, which becomes more marked posteriorly between the poorly developed chevron facets (Fig. 4C). The lateral sides of the centrum are strongly convex dorsoventrally and offset from the ventral surface by the broadly rounded edges of the ventral groove. On the dorsal part of the lateral side of the centrum, below the base of the neural arch, a notable, large pleurocentral depression is present (Fig. 4A). This depression is deeper posteriorly than anteriorly, with the anteroventral part of the depression forming a small lateroposteroventrally facing platform that is offset from the deeper posterior part by a rounded, but notable oblique step.

The walls of the neural arch are massive, and the neural canal is large (c. 35 mm in diametre) and round to slightly oval in outline. The base of the massive transverse process is placed entirely on the neural arch and extends for approximately the anterior three-fourths of the centrum. Posteriorly, the transverse process is supported ventrally by a stout, posterolaterally facing posterior centrodiapophyseal lamina, the ventral end of which does not reach the posterodorsal end of the centrum (Fig. 4B). Whereas the left lamina forms a sharp, posterolaterally facing edge, the right lamina seems to be more rounded, although this might be due to erosion. An anterior centrodiapophyseal lamina lamina is only indicated by a slight depression on the anterior side of the base of the transverse process. The transverse process was laterally and strongly posteriorly directed, but has almost no dorsal inclination. Posteriorly, a large postzygocentrodiapophyseal fossa
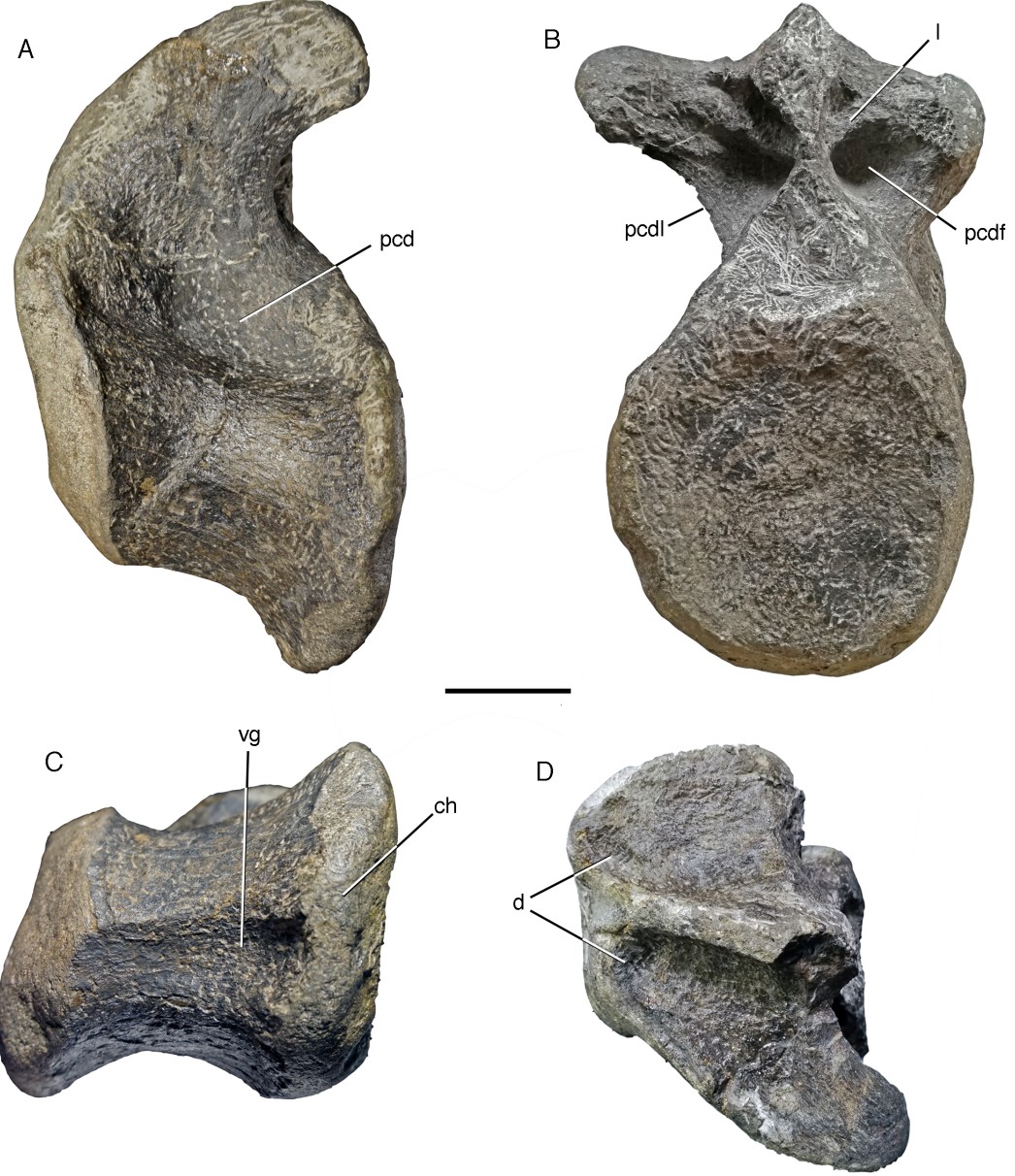

**Figure 4 Anterior caudal vertebra of a giant megalosaurid from the Vega Formation, MUJA-1913.**
(A) Left lateral view. (B) Posterior view. (C) Ventral view. (D) Dorsal view. Study sites: ch, chevron facet; d, depression on anterior end of dorsal surface of transverse process; l, lamina dividing the conical postzygocentrodiapophyseal fossa from a shallow dorsal depression; pcd, pleurocentral depression; pcdf, postzygocentrodiapophyseal fossa; pcdl, posterior centrodiapophyseal lamina; vg, ventral groove. Scale bar is 50 mm. Photographs by Oliver Rauhut and Diego Castanaera.

is present between the posterior centrodiapophyseal lamina and the lamina extending ventrally between the medial ends of the postzygapophyses and the neural canal (Fig. 4B); as the postzygapophyses are missing and the median lamina is poorly preserved, it is unclear if a small hyposphene might have been present, but at least a marked ventral expansion of this lamina was certainly absent. A small, ridge-like lamina extending
from the dorsal margin of the transverse process towards the dorsomedial rim of the neural canal subdivides the postzygocentrodiapophyeal fossa into a larger, conical ventral recess and a smaller, much shallower dorsomedial depression (Fig. 4B). Anteriorly, a small depression is present on the roof of the neural canal, being offset from the massive dorsal surface of the transverse process by a small, transverse step (Fig. 4D). The base of the broken neural spine is transversely narrow and extends over the entire length of the neural arch, showing the eroded bases of the slightly anteriorly diverging spinoprezygapophyseal laminae anteriorly.

## Asturian theropod tracks

Following the definition of *Marty et al. (2017)*, according to which giant theropod tracks are those of a footprint length (FL) longer than 50 cm, seven Asturian tracks are described in the present study (see Table 1 for measurements). The footprints (all more than 53 cm long), reported from the Kimmeridgian Lastres Formation, are preserved as natural sandstone casts and can be classified into two groups by morphology (*Piñuela Suárez, 2015*).

**Morphotype A** is represented by four tracks (Argüero1, Oles and Tazones specimens, and MUJA-1889; Fig. 5), which, although slightly different in morphology, are robust and weakly mesaxonic. The FL/footprint width (FW) ratio is very low (0.88–1.16). The digit impressions are broad and generally show claw marks. The divarication angle (II–IV) lies between 36° and 40°. In some of these tracks the digital pads are visible. Based on the morphology, the Asturian footprints would form part of the *Megalosauripus–Kayentapus*-group proposed by *Piñuela Suárez (2015)*, The specimens of morphotype A are thought to represent more graviportal theropods (*Piñuela Suárez, 2015*) than those of morphotype B.

**Argüero specimen 1.** The poorly preserved track represents a positive hyporelief. It is 70 cm in length and 62 cm in width; thus, the FL/FW ratio is very low (0.88), considerably lower than in the other tracks of the morphotype (Fig. 5A). The digit impressions are broad and relatively short, the best preserved being digits II and III. The claw marks are evident, well developed and medially turned. It is possible to recognize two pads in digit II. Digit IV is not well preserved, but enough is present to measure the divarication angle between digits II and IV, which is 36°. Even though the end of the digit IV is not preserved, the print seems weakly mesaxonic.

**Oles specimen.** The footprint represents a shallow positive epirelief (Fig. 5B). It is 82 cm in length and 66 cm in width, so the FL/FW ratio is 1.24. The digit impressions are broad, slight less so than in the previous specimen (Argüero specimen 1), and relatively short. Claw marks are evident in the three digits, being long and broad in digit II and shorter and narrower in III and IV. The digital pads are subtly visible, at least in digits III and IV. The divarication angle between digits II and IV is 38°. The track is weakly mesaxonic.

**Tazones specimen.** The print represents a positive hyporelief (Fig. 5C). It is 57 cm in length and more than 47 cm in width (the end of the digit IV is not preserved), so the FL/FW ratio is at least 1.21. The digit impressions are long and less broad than in the previous specimens. The claw marks, only preserved in digits II and III, are relatively large,

**Table 1 Measurements of the Asturian tracks.**

|  | Foot | FL | FW | FL/FW | II–IV |
|---|---|---|---|---|---|
| **Morphotype A** | | | | | |
| Argüero | R | 62 | 70 | 0.88 | 36 |
| Oles | L | 82 | 66 | 1.24 | 38 |
| Tazones | L | 57 | >47 | >1.16 | 38 |
| MUJA-1889 | L | 53 | 53 | 1 | 40 |
| **Morphotype B** | | | | | |
| MUJA-1263 | R | 62 | 38 | 1.63 | 15 |
| MUJA-0213 | R | 78 | | | |
| Argüero | R | 67 | | | |

**Notes:**
R, right foot; L, left foot; FL, footprint length; FW, footprint width; II–IV total divarication angle. For the specimens see Figs. 5 and 6.

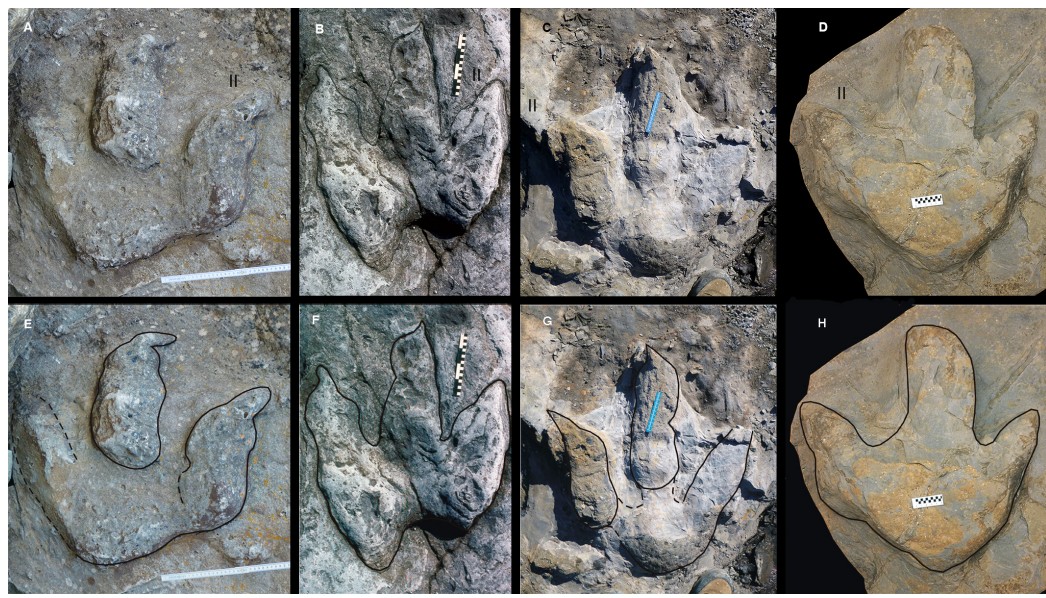

**Figure 5 Asturian Jurassic footprints with a weak mesaxony and probably related to very large or giant megalosaurid theropod trackmakers (Morphotype A).** (A), (B), and (C) specimens still on Argüero, Oles, and Tazones sea cliffs, respectively. Note that track (C) does not preserved the end of the digit IV. (D) MUJA-1889. (E–H) Same specimens, photographs with outline drawings to better illustrate track morphology. Photographs by José-Calros García-Ramos.

especially in digit II. The digital pads are subtly visible in digits II and III. The divarication angle between digits II and IV is 38°. Despite that the end of digit IV is not preserved, the print seems weakly mesaxonic. This footprint might be the best preserved of the morphotype A.

**MUJA-1889.** The track represents a positive hyporelief (Fig. 5D). It is 53 cm in length and 53 cm in width, so the FL/FW ratio is 1. The digit impressions are broad and short. The print is preserved as a shallow undertrack cast (associated to the true track cast), which might explain the poor definition of the claw marks, the absence of digital pads

and the relatively anterior position of the hypeces. The divarication angle between digits II and IV is apparently very high if taken from the undertrack (giving an incorrect value); using the true cast, the divarication angle (II–IV) is 40°. The print is also weakly mesaxonic. A horizontal outward translation movement is seen in this track, mainly in the digits II and III. The maximum depth for the track is 16 cm in the distal part of digit III. The specimen MUJA-1889 was recovered close to the Tazones specimen, and the composition and the thickness of the sandstone beds are similar in both, suggesting that they are derived from the same stratigraphic level. Keeping in mind that MUJA-1889 represents a different preservation (true track and shallow undertrack casts are associated) and is also affected by an oblique foot displacement, the morphology of this footprint does not reflect the foot anatomy of the producer, and thus could have been made by the same trackmacker that produced the Tazones specimen.

**Morphotype B** is represented by three footprints (MUJA-1263, MUJA-0213, and Argüero specimen 2; Fig. 6), which are much longer than wide and show a strong mesaxony. Pad impressions are only preserved in one specimen. The claw impressions vary from narrow and short to wide and long. The morphology of these footprints does not fit in large or giant known theropod ichnogenera, but rather with smaller ones characterized by a higher mesaxony. This set of tracks seems to represent more cursorial theropods (*Piñuela Suárez, 2015*) than morphotype A.

**MUJA-1263.** This true sandstone cast represents a positive hyporelief and is associated with a shallow undertrack (Fig. 6A). The print is much longer (62 cm) than wide (38 cm), so the FL/FW ratio is high (1.63). The digit impressions are relatively broad and long, and the claw marks are large. Even though this specimen is interpreted as an undertrack, it is possible to recognize two pads in digit II and three in digit III. The divarication angle (II–IV) is very low (15°). Although the end of digit II is not well preserved, the print is clearly highly mesaxonic. The maximum depth of the track is 10 cm in the distal part of digit III.

**MUJA-0213.** The track represents a positive hyporelief (Fig. 6B). The posterior part of the track is not well preserved, and although it is difficult to recognize the proximal margin, the footprint is much longer (78 cm) than wide (at least 35 cm, but digit IV is not complete). The impression of digit III is very long and digit II is relatively short, but both of them are broad, due to flattening processes *sensu Lockley & Xing (2015)*. The claw marks are short and narrow. Only two subtly visible pads are preserved in digit II. Digit IV is not complete, but enough to measure the divarication angle between digits II and IV, 34°. The print is highly mesaxonic, even though the digit IV is not complete.

**Argüero specimen 2.** The footprint (an epirelief) seems to be longer (67 cm) than wide (detailed measurements cannot be taken, because digit IV is not preserved) (Fig. 6C). The impressions of the digits are broad, and digit III is very long, whereas digit II is relatively short. The claw marks in both are long and narrow and medially directed in digit III. Digital pads are not recognizable in the digits. The interdigital angle between digits II and III is high (36°). The print seems to have been highly mesaxonic, although digit IV is not preserved.

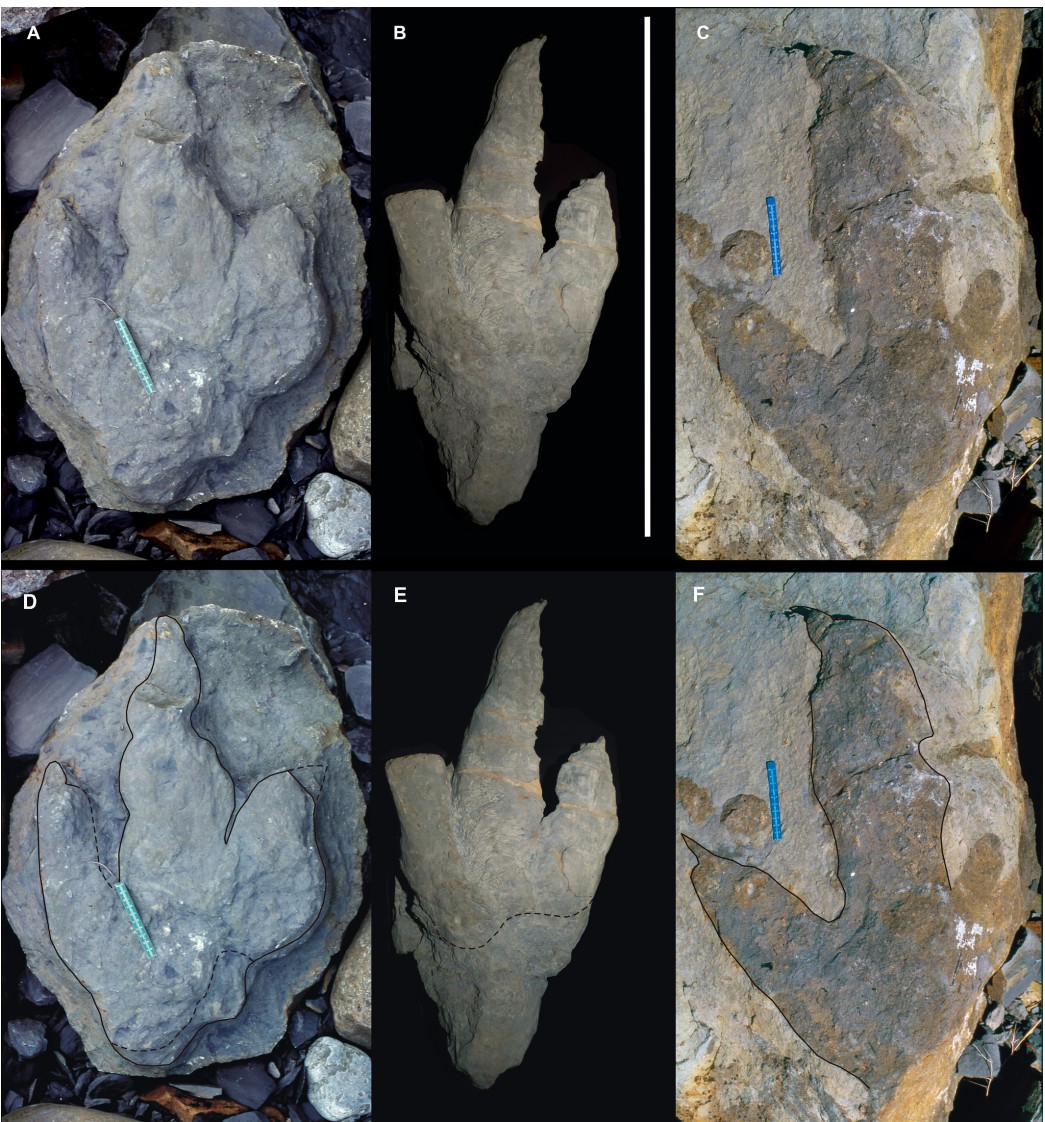

**Figure 6 Giant Asturian Jurassic footprints, strongly mesaxonic (Morphotype B).** (A) MUJA-1263. (B) MUJA-0213, scale bar: 1 m. (C) Specimen still on Argüero sea cliffs. (D–F) Same specimens, photographs with outline drawings to better illustrate track morphology. Photographs by José-Calros García-Ramos.

## DISCUSSION

### Systematic affinities of MUJA-1913

Despite the incomplete preservation of the caudal vertebra reported here, its systematic affinities can be narrowed down to at least a higher taxonomic category, although not to generic or species level. Large-bodied theropod dinosaurs reported from the Late Jurassic of Europe so far include members of the Ceratosauridae (*Antunes & Mateus, 2003*; *Mateus, Walen & Antunes, 2006*; *Malafaia et al., 2015*), Megalosauridae (*Antunes & Mateus, 2003*; *Mateus, Walen & Antunes, 2006*; *Carrano, Benson & Sampson, 2012*; *Hendrickx & Mateus, 2014*; *Malafaia et al., 2017a*), Metriacanthosauridae (*Huene, 1926*;

*Walker, 1964*; *Carrano, Benson & Sampson, 2012*), and Allosauridae (*Pérez-Moreno et al., 1999*; *Mateus, Walen & Antunes, 2006*; *Malafaia et al., 2007*, *2008a*, *2010*). Thus, comparisons of MUJA-1913 will mainly be with these clades.

Concerning the position of MUJA-1913 within the caudal vertebral column, this element can be quite confidently identified as a second or third caudal vertebra. Arguments in favour of this are the well-developed posterior centrodiapophyseal lamina, which is only present in the anteriormost caudal vertebrae, but is usually already less pronounced by caudal vertebra four (*Madsen, 1976*). On the other hand, the first caudal usually lacks chevron facets, but they are present on the posterior end of the centrum in MUJA-1913.

As no vertebral material of *Ceratosaurus* (the only ceratosaurian genus identified from the Jurassic of Europe so far) or any other ceratosaur has been described from the Jurassic of Europe, comparisons can only be made with *Ceratosaurus* from the Morrison Formation of the western US (*Gilmore, 1920*; *Madsen & Welles, 2000*). Anterior caudal vertebrae of this taxon differ from MUJA-1913 in the considerably higher than wide articular facets of the centrum (*Madsen & Welles, 2000*, pl. 7), the lack of a pronounced offset of the articular facets (*Gilmore, 1920*; *Madsen & Welles, 2000*), presence of a considerably narrower, deeper and better defined groove on the ventral side (*Gilmore, 1920*, p. 22; *Madsen, 1976*, Fig. 8B), the presence of a large, ventrally expanded hyposphene in the anterior caudals, and the relatively smaller and not subdivided postzygocentrodiapophyseal fossa (*Madsen & Welles, 2000*). The anterior caudal vertebrae of *Ceratosaurus* have marked pleurocentral depressions on the dorsolateral side of the centrum (see *Gilmore, 1920*, p. 22), but these are larger and less well-defined than in MUJA-1913. Concerning other ceratosaurian lineages, anterior caudal vertebrae of abelisaurs differ markedly from MUJA-1913 in lacking noted pleurocentral depressions, having well-developed hyposphenes in anterior caudals (with the exception of *Majungasaurus*; *O'Connor, 2007*), and usually strongly dorso-latero-posteriorly directed transverse processes (see *Méndez, 2014*). A referral of MUJA-1913 to Ceratosauria (*Martínez et al., 2000*; *Canudo & Ruiz-Omeñaca, 2003*) thus seems untenable.

The anterior caudal vertebrae of the metriacanthosaurids *Metriacanthosaurus* (OUMNH J 12144) and *Sinraptor* (IVPP 10600; *Currie & Zhao, 1993*) and the allosaurid *Allosaurus* (e.g. MOR 693; *Madsen, 1976*) have centra that are notably higher than wide, have less notably offset anterior and posterior articular facets, narrow towards their ventral side and lack both a notable pleurocentral depression on the lateral side of the centrum as well as the subdivision of the postzygocentrodiapophyseal fossa. Furthermore, a well-developed, ventrally expanded hyposphene is present in the anterior caudal vertebrae of metriacanthosaurids, and the ventral groove, if present, is notably narrower in allosauroids.

In contrast, the anterior caudal vertebrae of the megalosaurine megalosaurids *Megalosaurus* and *Torvosaurus* are very similar to MUJA-1913. Both of these taxa have very massive anterior caudal vertebral centra with a broad, posteriorly deepening ventral groove and a pronounced offset of the articular surfaces (NHMUK R 9672; BYU 13745; *Britt, 1991*; *Benson, 2010*), and the presence of marked pleurocentral

depressions on the lateral sides of the caudal centra was found to be a megalosaurine synapomorphy by *Rauhut, Hübner & Lanser (2016)*. Furthermore, these taxa lack expanded hyposphenes in the caudal vertebrae and a subdivision of the postzygocentrodiapophyseal fossa into a larger ventrolateral and a smaller, very shallow dorsomedial portion is also present in at least one vertebra of *Megalosaurus* (NHMUK R 9672), and seems to be also present in *Torvosaurus* (BYU 13745, BYU 5086). A small depression on the dorsal roof of the anterior part of the base of the transverse process, very similar to that in MUJA-1913, is also present in the anteriormost preserved caudal vertebra of the megalosaurid *Wiehenvenator* (*Rauhut, Hübner & Lanser, 2016*). Given these similarities, including the possibly apomorphic characters of marked pleurocentral depressions and a subdivided postzygocentrodiapophyseal fossa, we refer MUJA-1913 to an indeterminate megalosaurine megalosaurid. Given that the genus *Torvosaurus* has been identified from the Late Jurassic of the Iberian Peninsula (*Antunes & Mateus, 2003*; *Hendrickx & Mateus, 2014*; *Malafaia et al., 2017a*), this vertebra might represent this taxon, but a positive generic or specific identification of this incomplete element is impossible.

## Size of MUJA-1913

One striking feature of the vertebra from the Vega Formation is its enormous size. With a posterior centrum height of 150 mm, MUJA-1913 is larger than most anterior caudals for which measurements can be found in the literature. In particular, anterior caudals of *Torvosaurus tanneri* are about 25% smaller (*Britt, 1991*), an anterior caudal of *Spinosaurus aegyptiacus* is c. 10% smaller (*Stromer, 1915*), and one of the largest theropod caudals from the Jurassic, for which measurements were given, a possible carcharodontosaurid caudal from the Tendaguru Formation (*Rauhut, 2011*), is also c. 25% smaller than the specimen described here. Larger caudal vertebrae are present in the gigantic Cretaceous carcharodontosaurids (*Canale, Novas & Pol, 2015*) and *Tyrannosaurus* (*Brochu, 2003*), but might also be found in the largest allosauroid predators of the Late Jurassic Morrison Formation of the western USA (*Chure, 1995*, *2000*; *Williamson & Chure, 1996*), though no measurements are available in the literature for these specimens. However this may be, *Hendrickx & Mateus (2014)* argued that the holotype of *Torvosaurus guerneyi* represented the largest theropod dinosaur yet recorded from Europe (see also specimens described by *Malafaia et al., 2017a*). This specimen includes a partial anterior caudal vertebra, the posterior articular surface of which is about 15% smaller than that of MUJA-1913. Thus, given that the specimen from the Vega Formation probably belongs to a closely related taxon, this specimen probably represents the largest theropod dinosaur recorded so far in Europe, and represents an apex predator of more than 10 m in length.

It should be noted that *Pharisat (1993)* briefly reported large theropod caudal vertebrae from the Oxfordian of Plaimbois-du-Miroi, Doubs, France (see also *Allain & Pereda Suberbiola, 2003*), which, according to the measurements given, are of closely comparable size to MUJA-1913. Although no detailed description of these elements has ever been published, several characters indicate megalosaurid affinities for these elements:

the general shape of the centra and neural arches, the presence of a marked pleurocentral depression in the slightly more posterior vertebra, the almost circular outline of the posterior articular surface and the absence of a hyposphene in the probably first caudal, and the subdivision of the postzygocentrodiapophyseal fossa into a dorsomedial platform and a larger, conical ventrolateral depression (observations based on unpublished photographs provided by Daniel Marty and Christian Meyer; O. Rauhut, 2018, personal observations).

Other large Late Jurassic theropods from Europe have been reported on the basis of isolated teeth (*De Lapparent, 1943*; *Buffetaut & Martin, 1993*; *Rauhut & Kriwet, 1994*; *Canudo et al., 2006*; *Ruiz-Omeñaca et al., 2009c*; *Cobos et al., 2014*; *Gerke & Wings, 2016*; *Malafaia et al., 2017b*), and some of these specimens might represent animals that match MUJA-1913 in size (e.g. specimen described by *Cobos et al. (2014)*; largest specimens described by *Malafaia et al. (2017b)*). However, as relative tooth size varies widely in theropods, a direct size comparison is impossible.

## Ichnological evidence of giant theropods from the Kimmeridgian of Asturias

Regarding the giant theropod track record, *Cobos et al. (2014)* recently proposed that the Late Jurassic-Early Cretaceous (Berriasian) theropod tracks can be divided in two main groups (Ichno-group 1: *Bueckeburgichnus–Hispanosauropus–Megalosauripus* vs Ichno-group 2: *Iberosauripus*), which can be distinguished by their narrowness/ robustness, the proportion of the length of digit III (mesaxony) or footprint proportions (FL/FW ratio). The authors proposed that these two main groups might have been produced by members of Allosauridae and Megalosauridae, respectively.

We partially agree with the two ichno-groups related to the narrowness/robustness and strong/weak mesaxony proposed by *Cobos et al. (2014)* but less so with the ichnogenera included within them (due to unresolved problems in ichnotaxonomy), and the identification of some trackmakers (see below).

The validity of the Cretaceous German ichnogenus *Bueckeburgichnus Kuhn (1958)*, based on a poorly preserved footprint, is questionable, because the irregular shape of the digits and the relatively high total divarication angles suggesting extramorphological characters. Besides, the ichnogenus was created on the basis of only one specimen. Thus, the outline of the track reflects only partially the pedal morphology of the theropod. The tracks included in this ichnogenus were considered to be *Megalosauripus* by *Piñuela Suárez (2015*; see also *Hornung et al., 2012)*.

The same applies to *Hispanosauropus* (*Mensink & Mertmann, 1984*; *Lockley et al., 2007*) from the Kimmeridgian of Asturias, considered to be no valid ichnogenus by *Piñuela Suárez (2015)*, who included these Asturian tracks also in *Megalosauripus*. The poor preservation, which again does not reflect faithfully the foot morphology of the trackmaker, the probability of destruction and thus loss of the topotype located on an unstable sea cliff and the lack of a cast in any museum are enough reasons to reject the validity of this ichnogenus (see also *Lockley et al., 2007*).

Regarding *Megalosauripus*, this is the typical Late Jurassic-Early Cretaceous ichnotaxon in which many large theropod tracks have been included and that 'has often been used

as wastebasket in ichnotaxonomy' (see *Razzolini et al., 2017*; *Belvedere et al., 2018* and references therein).

The problem concerns the comparison between some *Megalosauripus* tracks with the recently defined large theropod ichnotaxa *Iberosauripus* (*Cobos et al., 2014*) or *Jurabrontes* (*Marty et al., 2017*).

On one hand, both shallow and deep undertracks belonging to large theropods, very frequent in Asturias and usually preserved as casts, are normally wider than the casts of the true tracks (*Piñuela Suárez, 2015*). This gives rise to footprints with relatively broader digit impressions, similar to *Megalosauripus uzbekistanicus* (type specimen of *Megalosauripus*), *M. teutonicus*, *Iberosauripus* or *Jurabrontes* (*Lockley, Meyer & Santos, 2000*; see also *Lockley et al., 1996*; *Diedrich, 2011*; *Cobos et al., 2014*; *Marty et al., 2017*).

On the other hand, tracks produced in carbonate sediments are often not well preserved. They sometimes tend, as in the undertracks, to be wider than the foot of the trackmaker and also show broader digit impressions. Moreover, according to *Razzolini et al. (2017)* the material of *Iberosauripus grandis* is rather poorly preserved. As stated correctly by *Dalla Vecchia (2008*, p. 99) 'the footprint morphology is highly influenced by the properties of the substrate, mainly in carbonate sedimentary settings' (see also *Dalla Vecchia & Tarlao, 2000*; *Belvedere et al., 2008*; *Fanti et al., 2013*). Thus, the substrate might have played a role when comparing large to giant theropod tracks, giving relatively similar footprint morphologies. Although some comparisons have recently been offered by *Marty et al. (2017)* and *Razzolini et al. (2017)*, a detailed revision of the ichnogenus *Megalosauripus*, including the three different ichnospecies (*M. uzbekistanicus, M. teutonicus*, and *M. transjuranicus*), and an evaluation of the possible impact of locomotion and substrate in the production and distinction of large to giant theropod tracks, such as *Iberosauripus* and *Jurabrontes* (*Marty et al., 2017*) are necessary to clarify the ichnotaxonomic status of the Asturian tracks. In this respect, it is noteworthy that some of the Asturian tracks (Argüero specimen 1, Oles specimen) of morphotype A described here generally resemble *Jurabrontes*, as described by *Marty et al. (2017)*. Nonetheless, they are also similar to *M. uzbekistanicus* and *M. teutonicus* (*Lockley et al., 1996*; *Lockley, Meyer & Santos, 2000*), to some Late Jurassic tracks assigned to *Megalosauripus* isp. (*Diedrich, 2011*; *Lallensack et al., 2015*; *Mazin, Hantzpergue & Pouech, 2016*; *Mazin, Hantzpergue & Olivier, 2017*) and to *Iberosauripus* (*Cobos et al., 2014*). Some specimens of morphotype B resemble tracks also assigned to *Megalosauripus*, but to the recently defined ichnospecies *M. transjuranicus* (*Razzolini et al., 2017*), characterized by a higher mesaxony and its gracility in comparison with the other aforementioned tracks, although this ichnospecies never reached the size of the Asturian specimens.

Following the previous considerations, and given the poor preservation and the ichnotaxonomical problems with the large to giant theropod tracks, we tentatively consider the Asturian morphotype A as *Megalosauripus*-like, while the Asturian morphotype B cannot be classified within any known ichnotaxa. The notably divergent morphology of the tracks included in morphotypes A and B indicates that at least two taxa of giant theropod were present in the Kimmerdigian of Asturias, as it seems very

unlikely that the marked difference in mesaxony between these morphotypes can be attributed to differences in preservation. In general terms, the two morphotypes conform to the distinction proposed by *Cobos et al. (2014)* in that morphotype A represents a very robust animal, whereas morphotype B seems to stem from a more gracile theropods. The presence of two large theropods, one gracile and one robust, has already been described in other Late Jurassic areas, such as the Jura Carbonate platform (*Jurabrontes curtedulensis* and *M. transjuranicus*, *Razzolini et al., 2017*; *Marty et al., 2017*) or the Iouaridène Formation in Morocco (*Megalosauripus* and unnamed giant theropod tracks, *Boutakiout et al., 2009*; *Belvedere, Mietto & Ishigaki, 2010*).

With up to 82 cm, the Asturian specimens show FLs that fall within the range of the largest tracks in the world (*Boutakiout et al., 2009*; *Piñuela Suárez, 2015*; *Marty et al., 2017*). Some of these large predators from the Late Jurassic of Asturias apparently had cursorial adaptations, as deduced from the morphological study of their footprints (morphotype B), which show strong mesaxony (*sensu Lockley, 2009*); their claw impressions, when preserved, are long and very narrow. These dinosaurs were as large as, but more agile than trackmakers of Morphotype A tracks. The largest theropod trackmakers from the Jurassic of Asturias were thus similar in size to *Tyrannosaurus rex*, based on known footprints of that taxon (*Lockley & Hunt, 1994*; *Manning, Ott & Falkingham, 2008*; *McCrea et al., 2014*) and foot skeletons (*Brochu, 2003*).

## Late Jurassic apex predators in Europe

Apart from the ichnotaxonomic questions discussed above, the question remains which theropod groups are represented by these giant tracks. As noted above, *Cobos et al. (2014)* suggested a division of theropod tracks into two larger categories of robust and gracile prints (regardless of the exact identification to ichnogenus or ichnospecies level), which they considered to represent megalosaurids and allosaurids, respectively. The main argument for this identification was the relative robustness or slenderness of the tracks, as the only well-known Late Jurassic megalosaurid, *Torvosaurus*, is a very robust animal (*Britt, 1991*; *Hendrickx & Mateus, 2014*; *Malafaia et al., 2017a*), whereas the best known allosaurid, *Allosaurus*, is much more gracile (*Gilmore, 1920*). Consequently, *Cobos et al. (2014*, p. 37–38*)* argued that the more robust tracks were probably made by megalosaurids, whereas the more slender tracks correspond to allosaurids.

However, this suggestion is somewhat simplistic and problematic for several reasons. The first and obvious problem (also noted by *Cobos et al., 2014*) is that no complete pes is known in any large ceratosaurian or Jurassic non-coelurosaurian tetanuran, with the exception of *Allosaurus* (*Madsen, 1976*) and a specimen from the Lusitanian Basin that was originally also referred to *Allosaurus* (*Malafaia et al., 2008a*), but might represent a carcharodontosaur (*Malafaia et al., 2017c*). Even in the very complete holotype specimen of the metriacanthosaurid *Sinraptor dongi*, several pedal phalanges are missing (*Currie & Zhao, 1993*), and at the most isolated phalanges are known for megalosaurids (*Sereno et al., 1994*; *Allain & Chure, 2002*; *Sadleir, Barrett & Powell, 2008*). Thus, a synapomorphy-based correlation (sensu *Carrano & Wilson, 2001*) between pedal morphology and trackways in large basal tetanurans is currently impossible.

However, known complete pedes of *Allosaurus* (*Gilmore, 1920*; *Evers, 2014*) do not seem to show the extreme differences in digit III as opposed to digits II and IV that would lead to the mesaxony seen in one of the largest footprints ascribed to morphotype B described here (MUJA-0213). This extreme mesaxony is a strange situation in large theropod tracks as generally they tend to show lower mesaxony values than smaller theropod tracks (e.g. *Grallator-Eubrontes* plexus; *Lockley, 2009*).

A second problem in the identification proposed by *Cobos et al. (2014)* is that it neither takes the systematic nor the morphological variation of known Jurassic averostrans that reach large to giant sizes into account. First, allosaurids are not the only large allosauroids known from Europe, with the English metriacanthosaurid *Metriacanthosaurus* representing an animal of similar or even greater size than known specimens of *Allosaurus* from Europe (*Huene, 1926*; *Walker, 1964*; *Pérez-Moreno et al., 1999*; *Mateus, Walen & Antunes, 2006*; *Malafaia et al., 2010*). However, the better known metriacanthosaurids from China are similar in proportions and robustness to *Allosaurus* (*Dong, Zhou & Zhang, 1983*; *Currie & Zhao, 1993*; *Gao, 1999*), and the pes of *Sinraptor* does also not seem to be significantly different from that of *Allosaurus* (see *Madsen, 1976*; *Currie & Zhao, 1993*). Thus, the more slender tracks of Ichno-Group 1 of *Cobos et al. (2014)* might represent metriacanthosaurids as well as allosaurids. On the other hand, the largest allosaurid known from the Late Jurassic Morrison Formation of North America, *Saurophaganax*, is a more robustly built animal (*Chure, 1995*, *2000*), whereas afrovenatorine megalosaurids, such as *Afrovenator* (*Sereno et al., 1994*) and *Eustreptospondylus* (*Sadleir, Barrett & Powell, 2008*) are rather gracile animals. Although *Eustreptospondylus* from the Callovian–Oxfordian boundary of England represents the youngest afrovenatorine currently known from Europe (and, possibly globally, depending on the uncertain age of *Afrovenator*), the Late Jurassic European theropod fossil record is insufficient to completely rule out their survival into later stages, and at least caution is advisable in identifying tracks as allosauroid on the basis of their slenderness only.

Finally, the basal ceratosaur *Ceratosaurus*, known from the Late Jurassic of Portugal (*Antunes & Mateus, 2003*; *Mateus, Walen & Antunes, 2006*; *Malafaia et al., 2015*) is a rather large animal as well (*Gilmore, 1920*; *Madsen & Welles, 2000*). Although the holotype of *Ceratosaurus nasicornis* has been estimated with a total length of slightly more than 5 m (*Gilmore, 1920*), the type of *C. dentisulcatus* is about 22% larger (*Madsen & Welles, 2000*), and other specimens (e.g. BYU 881) reach sizes comparable to that of large specimens of *Allosaurus*. As *Ceratosaurus* is also a rather gracile animal, exceptionally large individuals of this or a closely related taxon could also have made the more gracile tracks.

Concerning megalosaurine megalosaurids, no pedal elements other than metatarsals have been described for any of the included genera *Duriavenator*, *Megalosaurus*, *Wiehenvenator*, and *Torvosaurus* (*Galton & Jensen, 1979*; *Britt, 1991*; *Benson, 2008b*, *2010*; *Hanson & Makovicky, 2014*; *Hendrickx & Mateus, 2014*; *Rauhut, Hübner & Lanser, 2016*; *Malafaia et al., 2017a*). However, at least *Megalosaurus*, *Wiehenvenator*, and *Torvosaurus* are notable for their extreme robustness, and *Williamson & Chure (1996*, p. 78) cite a personal communication by James Madsen, according to which the pedal phalanges of *Torvosaurus* are notably short and wide. These observations are thus in agreement

with the suggestion by *Cobos et al. (2014)* that the very robust tracks with a low mesaxony might represent (megalosaurine) megalosaurids. Nevertheless, we agree with *Marty et al. (2017)* that caution is advisable in assigning giant theropod tracks from the Jurassic to any clade unless better data on pedal morphology in basal tetanurans becomes available.

Regardless of the exact identification of the trackmaker, European sites have yielded some of the largest known Jurassic theropod tracks, such as the trackways described from the Middle Jurassic of Oxforshire, UK, (*Day et al., 2004*) and Vale de Meios, Portugal, (*Razzolini et al., 2016*), which were made by giant theropods, tentatively attributed to *Megalosaurus* and to the Megalosauridae, respectively. Recently, *Marty et al. (2017)* described new giant theropod tracks (*J. curtedulensis*) from the Kimmeridgian of NW Switzerland. This new ichnotaxon is characterized by tracks that are slightly longer than wide and show weak mesaxony, and, as the authors suggested, can be included within the main features of the Ichno-Group 2 of *Cobos et al. (2014)*. These authors emphasized that some of the *Jurabrontes* tracks are among the largest theropod tracks worldwide. However, the Kimmeridgian of Asturias is the only Jurassic European site that has yielded tracks of two giant theropods (gracile and robust) so far, indicating that two different clades of giant theropods were present here.

Concerning osteological remains, the identification of MUJA-1913 as a megalosaurid adds to the already diverse European fossil record of the clade. As discussed by *Benson (2010)*, *Carrano, Benson & Sampson (2012)* and *Rauhut, Hübner & Lanser (2016)*, megalosaurids were taxonomically diverse and widespread in the Middle Jurassic of Europe. However, whereas megalosaurids are rare in the Kimmeridgian–Tithonian Morrison Formation of the western US (*Foster, 2003*; *Rauhut, Hübner & Lanser, 2016*), and unknown from the Late Jurassic of Asia, they seem to be abundant and wide-spread in the Late Jurassic of Europe. From the Lusitanian Basin, the large megalosaurid *Torvosaurus gurneyi* and several other megalosaurid postcranial specimens, numerous teeth, as well as eggs and embryos were described (*Antunes & Mateus, 2003*; *Mateus, Walen & Antunes, 2006*; *Malafaia et al., 2008b, 2017a, 2017b*; *Araújo et al., 2013*; *Hendrickx & Mateus, 2014*). From the Late Jurassic Villar del Arzobispo Formation of the Iberian Range, *Gascó et al. (2012)* and *Cobos et al. (2014)* referred isolated teeth to the Megalosauridae, including the largest tooth specimen found in these rocks (*Cobos et al., 2014*). Likewise, *Gerke & Wings (2016)* identified the largest theropod teeth in their sample from the Kimmeridgian of northern Germany as probable megalosaurids. Furthermore, the early juvenile megalosaurid *Sciurumimus* was found in the Kimmeridgian Torleite Formation of Bavaria (*Rauhut et al., 2012*; the layers were referred to the Rögling Formation in that paper, but recent lithostratigraphic revisions place the Kimmeridgian beds at Painten in the Torleite Formation; *Niebuhr & Pürner, 2014*). Apart from the fragmentary skeleton of the large-bodied metriacanthosaurid *Metriacanthosaurus* from the Oxfordian Oxford Clay (*Huene, 1926*; *Walker, 1964*), all identifiable large theropod remains from the Late Jurassic of England seem to represent megalosaurids as well, including remains of a large maxilla and a very robust tibia from the Kimmeridge Clay (*Benson & Barrett, 2009*; *Carrano, Benson & Sampson, 2012*).

As noted above, the largest Jurassic theropod remains found in France (*Pharisat, 1993*) also seem to represent a megalosaurid. The specimen described here from the Kimmeridgian of Asturias fits well in this general panorama.

Thus, megalosaurid theropods seem to have represented the largest predators on all of the Late Jurassic European landmasses that we have fossil evidence for, together with allosaurids in the western parts of Europe and metriacanthosaurids in the eastern areas. As these parts of Europe were an assemblage of medium-sized islands and most of the sediments that have yielded theropod remains are either nearshore terrestrial or even marine beds, this seems to support the suggestion of *Rauhut, Hübner & Lanser (2016)* that megalosaurids might have preferred nearshore environments, and that the apparent faunal change from megalosaurid-dominated to allosauroid-dominated faunas from the Middle to the Late Jurassic might rather reflect regional and environmental biases in the fossil record of Jurassic theropods.

Given the abundance and wide distribution of megalosaurids in the Late Jurassic of Europe, the question arises if different lineages of megalosaurids populated the different landmasses, possibly evolving in isolation from their Middle Jurassic predecessors, or if an interchange of megalosaurid taxa between the different islands might have been possible. The presence of abundant theropod tracks, the largest of which are often related to megalosaurids, in shallow marine or carbonate platform environments (*Marty et al., 2017*) might indicate that short time sea level changes may have allowed some faunal interchange between otherwise separate landmasses during the Late Jurassic (*Meyer, 2012*). Indeed, *Marty et al. (2017)* suggested that the Jura carbonate platform could have represented a 'faunal exchange corridor' of the dinosaur faunas between the southern and the northern landmasses.

Similarly large theropod tracks have also been reported from the Late Jurassic of northern Africa (*Boutakiout et al., 2009*). *Belvedere (2008*; see also *Belvedere, Mietto & Ishigaki, 2010*; *Marty et al., 2010*) noted great similarities between ichnofaunas from the Late Jurassic of Morocco and the Jura Mountains. The possibility of faunal interchange between Europe and North Africa through an Iberian corridor during the Early Cretaceous was discussed by *Canudo et al. (2009)*, who concluded that such an interchange was improbable before the Barremian–Aptian. In the Late Jurassic, at least along the south–south-eastern margin of Iberia, this land mass was separated from Africa by oceanic floor (*Olóriz, 2002*), and sediments from the northern shore of this oceanic basin in the Betic Cordillera indicate pelagic conditions (*Olóriz et al., 2002*), indicating that there was a rather wide separation of Iberia from northern Africa in this region. Even though the Ligurian sea floor spreading most probably did not extend into the region of the opening central Atlantic (*Ford & Golonka, 2003*), continental rifting extended between the Ligurian ocean and the area around Gibraltar, forming a considerably thinned continental lithosphere consisting of pull-apart basins that make up the Alboran Basin, which, with a width of at least 100–200 km (and possible twice as much), separated northern Africa from Iberia during the Late Jurassic (*Capitanio & Goes, 2006*), being flooded by epicontinental seas. This region furthermore experienced significant transformational tectonics (*Capitanio & Goes, 2006*). Although

short time emergence of parts of this area due to eustatic sea level changes might not be completely impossible, the complete formation of a land bridge between Iberia and northern Africa in the Late Jurassic seems unlikely. Although at least sporadic intervals of faunal interchange cannot be completely ruled out, the possibility of a dinosaur interchange between Europe and northern Africa during the Late Jurassic seems rather improbable due to the continuous seaway (the 'Hispanic Corridor') connecting the Tethys sea with the Panthalassan ocean, as revealed by known palaeogeographic reconstructions (*Ziegler, 1988*; *Dercourt et al., 2000*; *Ford & Golonka, 2003*; *Vrielynck & Bouysse, 2003*). In addition, the global sea-level reached its Jurassic maximum during the Late Kimmeridgian-Early Tithonian times, although short-time fluctuations in sea level are also notable during this interval (*Haq, 2018*).

Unfortunately, there is no osteological record of theropods from the Late Jurassic of northern Africa, so nothing can be said about possible faunal similarities and differences between this region and Europe. Traditionally, scientists have pointed out the allegedly great similarity of the Late Jurassic fauna of the east African Tendaguru Formation to that of the Morrison Formation (*Galton, 1977*, *1982*) and the Lusitanian Basin (*Mateus, 2006*), but recent research has rather emphasized the differences between this Gondwanan fauna from its contemporaneous Laurasian counterparts (*Remes, 2006*; *Taylor, 2009*; *Hübner & Rauhut, 2010*; *Rauhut, 2011*). Interestingly, though, the Tendaguru theropod fauna seems to also include at least three large to giant theropod taxa, including a possible abelisaurid, a possible megalosaur, and a probably carcharodontosaurian allosauroid (*Rauhut, 2011*). Thus, the same general lineages are present in the fauna of apex predators in eastern Africa and Europe, although the exact clades represented might be different (though note that *Malafaia et al. (2017c)* recently identified the first possible carcharodontosaurian from the Lusitanian Basin). Whether this is due to shared heritage from Pangean times, or if some faunal interchange might, at least sporadically, have been possible can only be answered in the light of future discoveries from northern Africa.

On the other hand, the comparison between theropod tracks of both continental blocks might not be too significant, since, as pointed out by *Farlow (2001*, p. 417–421*)*: '. . . pedal phalangeal skeletons of large ceratosaurs, allosaurs, and tyrannosaurs are indistinguishable. That being the case, it is probably impossible to correlate large-theropod footprints with the clades of their makers on the basis of print shape alone . . . using large-theropod ichnotaxa to make intercontinental correlations (. . .) is a procedure that should be done with considerable caution. Footprints that on morphological grounds can be placed in the same ichnotaxon might have been made by large theropods that were not closely related.'

However, the different features seen in the large theropod tracks from Asturias, the Jura carbonate platform and Morocco in the Late Jurassic seem to at least partially contradict *Farlow (2001)*. The presence of two different large to giant theropods in the Late Jurassic is supported by the ichnological evidence in several places, in which clearly distinguishable robust and gracile morphotypes are found, suggesting that, although

different genera/species might have inhabited Europe and North Africa, two groups, one gracile, and one robust, were present.

## CONCLUSIONS

The presence of very large theropods in the Asturian Basin (Northern Spain) during the Upper Jurassic (Kimmeridgian) is confirmed by both the footprints and skeletal remains. Whereas the only skeletal remain of a giant theropod from the Vega Formation represents a megalosaurine megalosaurid, the track record indicates at least two taxa of giant theropods in the slightly younger Lastres Formation. Both osteological and ichnological evidence indicates that very large to giant theropod dinosaurs were widespread in Europe in the Late Jurassic, and the largest representatives seem to have been close to the maximum body size recorded for theropods. Given that Europe represented an assemblage of larger and smaller islands at that time, this is surprising, as maximum body size is usually correlated with available land mass in vertebrates (*Marquet & Taper, 1998*; *Burness, Diamond & Flannery, 2001*), and island dwarfing has been reported in dinosaurs (*Sander et al., 2006*; *Stein et al., 2010*). A possible solution to this apparent contradiction might be that short time sea level changes allowed faunal interchange between the different islands that constituted Europe repeatedly during the Late Jurassic. Dinosaur tracks preserved in shallow marine carbonate platform environments might be direct evidence for this (*Marty et al., 2017*). The preference of nearshore environments in megalosaurids, possibly in search for suitable food (*Razzolini et al., 2016*) might furthermore explain the wide distribution of this group in the European archipelago.

## INSTITUTIONAL ABBREVIATIONS

| | |
|---|---|
| **BYU** | Brigham Young University, Provo, USA |
| **IVPP** | Institute of Vertebrate Paleontology and Paleoanthropology, Beijing, China |
| **MOR** | Musem of the Rockies, Bozeman, USA |
| **MUJA** | Museo del Jurásico de Asturias, Colunga, Spain |
| **NHMUK** | Natural History Museum, London, UK |
| **OUMNH** | Oxford University Museum of Natural History, Oxford, UK. |

## ACKNOWLEDGEMENTS

We thank Daniel Marty and Christian Meyer for providing photographs of the theropod vertebrae from Plaimbois-du-Miroi and help with literature and Adriana López-Arbarello for discussions. Matteo Belvedere and Elisabete Malafaia improved the work with critical comments.

### Funding

This work was supported by the Deutsche Forschungsgemeinschaft (DFG) under grant RA 1012/23-1 (to Oliver Rauhut) and by the Ministerio de Economia y Competitividad

(MINECO) and Fondo Europeo de Desarrollo Regional (FEDER) under grant CGL2015-66835-P (to Laura Piñuela). Diego Castanera was supported by a postdoctoral fellowship of the Humboldt Fundation. The funders had no role in study design, data collection and analysis, decision to publish, or preparation of the manuscript.

## Grant Disclosures

The following grant information was disclosed by the authors:
Deutsche Forschungsgemeinschaft (DFG): RA 1012/23-1.
Ministerio de Economia y Competitividad (MINECO).
Fondo Europeo de Desarrollo Regional (FEDER): CGL2015-66835-P.

## Competing Interests

The authors declare that they have no competing interests.

## Author Contributions

- Oliver W.M. Rauhut conceived and designed the experiments, performed the experiments, analyzed the data, prepared figures and/or tables, authored or reviewed drafts of the paper, approved the final draft.
- Laura Piñuela conceived and designed the experiments, performed the experiments, analyzed the data, contributed reagents/materials/analysis tools, prepared figures and/or tables, authored or reviewed drafts of the paper, approved the final draft.
- Diego Castanera conceived and designed the experiments, performed the experiments, analyzed the data, authored or reviewed drafts of the paper, approved the final draft.
- José-Carlos García-Ramos conceived and designed the experiments, performed the experiments, analyzed the data, contributed reagents/materials/analysis tools, prepared figures and/or tables, authored or reviewed drafts of the paper, approved the final draft.
- Irene Sánchez Cela performed the experiments, contributed reagents/materials/analysis tools, authored or reviewed drafts of the paper, approved the final draft, preparation of materials.

## Data Availability

The specimens described are stored in the Museo del Jurásico de Asturias (MUJA), if not indicated otherwise (see institutional abbreviations in the manuscript).

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
