# Peer review of "The largest European theropod dinosaurs: remains of a gigantic megalosaurid and giant theropod tracks from the Kimmeridgian of Asturias, Spain"

_PeerJ, doi:10.7717/peerj.4963_

## Round 0.1 · original submission · Major Revisions

· Academic Editor

Major Revisions

I concur with the reviewers that the work is a good contribution to dinosaur paleobiology. I also agree with them though that the study would greatly benefit from additional information, not only for the discussion of known material from the Jurassic (proposed by the second reviewer), but by adding more graphic and morphometric data to substantiate descriptions, as proposed by the first reviewer. In effect, the suggestion of including the Moroccan record is definitely logical, given the paleobiogeographical affinities, and the concerns about taxonomic precision and designation of the tracks could be briefly discussed too. Furthermore, to aid descriptions I would also strongly encourage the authors not only to graphically outline the material, but also to include 3D models which, in my own experience, can be easily and affordably attained, specially from tracks, using photogrametry.

·

Basic reporting

The conclusions are consistent with the data provided, although I suggest to expand the comparison record also to the North African (e.g. Moroccan) bone and track record.
I would also add more details on the tracks presented, with more deteiled descritpions and a more complete table.
Figures are all nice and very accurate, but I'd add some outline drawings for the tracks, and if possible some 3D models, to help the readred understating the author's interpretation.

Experimental design

no comment

Validity of the findings

no comment

Additional comments

The manuscript is sound and worth the publication after major (moderate) revisions.
Specifically, I think that a more deteiled description and thorough interpretation of the tracks is needed, both for more solid interpratation and to provide the reader with enough information on the tracks. Specifically, I find the table too reduced, and I suggest to add more fundamental measurements for all the tracks examined.

I agree with the division of the two types, but I have some doubt about the interpretation, which seems to be quite vague, despite the good quality of the material presented. In my opinion tracks of type A are very similar enough to Jurabrontes and Iberosauripus, while the tybe B (In particular Fig 6A) are very similar to Megalosauripus, as defined in Razzolini et al (2017).
Some interpratative outline drawings should be added to help the reader understanding how the interpretation was made, and to prevent misinterpretation given by the lightening of the tracks (especially of those in the field). Also, if possible, some 3D models should be added to the publciation.

More in general, it owuld also be interesting to support your temporary connection hypothesis, a comparison with the bone and track record from N. Africa.
Detailed comments are in the pdf.

I hope my comments will help you improving the manuscript. Feel free to contact me to discuss about any comment I have made.

Best regards

·

Basic reporting

No comment

Experimental design

No comment

Validity of the findings

No comment

Additional comments

The authors describe an interesting set of theropod specimens from Spain, which adds to the scarce information currently available about the Spanish record of Upper Jurassic theropods. The work represents an increase to the known Iberian record of theropods and has great interest for knowing Iberian Late Jurassic theropod faunas and for understanding their biogeography.

I have only a few questions and punctual suggestions:

The authors say that no complete pes is known in any Jurassic non-coelurosaurian tetanuran with the exception of Allosaurus. An almost complete pes of an allosauroid from the Portuguese Upper Jurassic, first interpreted as an Allosaurus and more recently tentatively assigned to a carcharosontosaurian allosauroid (Malafaia et al. 2008; 2017) could add some information to the discussion regarding the possible identification of the different track morphotypes. A detailed description of the specimen is not available and its identification is unstable, but as the most complete theropod pes currently known in the Upper Jurassic Iberian record, I think it could be useful to mention it.

In Geological setting
L. 134-135. Ruiz-Omeñaca et al. 2009c reported a second tooth (MUJA-1018) from the same locality (Playa da Vega). Could be this specimen also from the same level as the theropod caudal vertebra?
L. 141-143. Are the sauropod caudal vertebrae and the quadrupedal dinosaur footprints unpublished or are those mentioned in Ruiz-Omeñaca et al. 2006 and Lockley et al. 2008?

In Size of MUJA-1913
L. 311-318. The last sentence of the second paragraph is very long. I suggest number the features to make it easier to read.

In Late Jurassic apex predators in Europe
L. 403. Instead of "pedes" should not be "pes"?

---

## Round 0.2 · accepted · Accept

· Academic Editor

Accept

The reviewers agree that the manuscript has been greatly improved, and so do I, and though I also think that a 3D digital model would have been a great addition, the ms is good in its present form.

# ·

Basic reporting

no comment

Experimental design

no comment

Validity of the findings

no comment

Additional comments

The manuscript has improved and is suitable for publication without further modification.
The comments to my remarks are sound and the text ahs been modified accordingly.
It's is a pity that no 3D models could be added, but I understand the logistic issues and hope that they will be published in the future.